An automated AI-powered IoT algorithm with data processing and noise elimination for plant monitoring and actuating

Ali Mohammed A. H. 1 2 hashem@um.edu.my
Moiduddin Khaja 3
Nukman Yusoff 2
Abd Razak Bushroa 1 2
Aboudaif Mohamed K. 3
http://orcid.org/0000-0001-5487-545X Thangaraj Muthuramalingam 4
1 Advanced Manufacturing and Materials Processing (AMMP), Faculty of Engineering, Universiti Malaya , Kuala Lumpur , Malaysia
2 Department of Mechanical Engineering, Faculty of Engineering, University of Malaya , Kuala Lumpur , Malaysia
3 Advanced Manufacturing Institute, King Saud University , Riyadh , Saudi Arabia
4 Department of Mechatronics Engineering, Faculty of Engineering and Technology, SRM Institute of Science and Technology , Chennai , India
Alatas Bilal
Electronic publication date: 2024 Nov 6
Publication date: 2024
Volume: 10
Electronic Location ID: e2448
Received 2024 Jun 21; Accepted 2024 Oct 4
Copyright: © 2024 Ali et al.
Copyright year: 2024
Copyright holder: Ali et al.
License: This is an open access article distributed under the terms of the Creative Commons Attribution License, which permits unrestricted use, distribution, reproduction and adaptation in any medium and for any purpose provided that it is properly attributed. For attribution, the original author(s), title, publication source (PeerJ Computer Science) and either DOI or URL of the article must be cited.
License URL: https://creativecommons.org/licenses/by/4.0/

Keywords: Automated AI-Powered IoT farming, Enhanced Laser Simulator Logic (ELSL), IoT monitoring, Image and signal processing, CO2 detector and soil moisture sensors, Chili

Funding: King Saud University, Riyadh, Saudi Arabia RSPD2024R1076 This research was funded through the Researchers Supporting Project number (RSPD2024R1076), King Saud University, Riyadh, Saudi Arabia. The funders had no role in study design, data collection and analysis, decision to publish, or preparation of the manuscript.

==============================
This article aims to develop a novel Artificial Intelligence-powered Internet of Things (AI-powered IoT) system that can automatically monitor the conditions of the plant (crop) and apply the necessary action without human interaction. The system can remotely send a report on the plant conditions to the farmers through IoT, enabling them for tracking the healthiness of plants. Chili plant has been selected to test the proposed AI-powered IoT monitoring and actuating system as it is so sensitive to the soil moisture, weather changes and can be attacked by several types of diseases. The structure of the proposed system is passed through five main stages, namely, AI-powered IoT system design, prototype fabrication, signal and image processing, noise elimination and proposed system testing. The prototype for monitoring is equipped with multiple sensors, namely, soil moisture, carbon dioxide (CO2) detector, temperature, and camera sensors, which are utilized to continuously monitor the conditions of the plant. Several signal and image processing operations have been applied on the acquired sensors data to prepare them for further post-processing stage. In the post processing step, a new AI based noise elimination algorithm has been introduced to eliminate the noise in the images and take the right actions which are performed using actuators such as pumps, fans to make the necessary actions. The experimental results show that the prototype is functioning well with the proposed AI-powered IoT algorithm, where the water pump, exhausted fan and pesticide pump are actuated when the sensors detect a low moisture level, high CO2 concentration level, and video processing-based pests’ detection, respectively. The results also show that the algorithm is capable to detect the pests on the leaves with 75% successful rate.

Introduction

Agriculture is a key sector that has a great impact on the people live, countries’ economic and the surrounding environment. The world’s population is expected to be grown from current 7.5 billion to 10 billion people in 2050 which indicates that there is a need for sustainable food and agriculture resources. There are around 570 million farms worldwide that need millions of agriculture workers to keep tracking on the crops and ensure the healthiness of the plants. In fact, the farmers cannot work in the field for 24 h and detect all harmful effects on the plants which need an automated based detection system (Abdullah et al., 2021). Up to date, the worldwide agricultural sector is relatively slow in adopting the modern technology to replace the traditional farming methods which are slow, time consuming and difficulty to continuously monitor the crops (Abdullah et al., 2021). The farmers may face various challenges such as sticking to the watering schedule and late detection of disease which are resulted in harming the plants by diseases, and lacking of CO2 concentration that affect to the plant growth (Mohamood et al., 2019).

There exit several automation and smart plant monitoring systems which do not fulfill the needs of customers and apply automatically the suitable actions upon detection of bad plant conditions (Li et al., 2018).

The current technological solutions for the plant monitoring system in literature are reviewed as follows:

Li et al. (2018) have reviewed most of existed automatic greenhouse system and concluded that the potential plant diseases do not incorporate well with automatic irrigation or a mechanism to regulate carbon dioxide level in the air and vice versa. The current smart monitoring systems for the plants need specialized persons to use them for remotely farming and tracking purposes (Vaidya, Ambad & Bhosle, 2018; Sharon et al., 2021), due to the complexity and the cost of existing systems.

Gultom et al. (2017) have developed automatic water monitoring system with Internet of Things (IoT) technology for chili product in Indonesia. It is equipped with a moisture sensor and Arduino microcontroller to detect the status of soil and determine the humidity, which are used to open or close the water pump valve. The system uses pH and EC sensors for detecting the acidity of chili plants and nutrient of the soil, respectively. The result was sent to the Web App using Ethernet Shield to enable the farmer for monitoring the chili plant in real time. An automated Internet of Things (IoT) based water monitoring has been developed by Hanumann et al. (2022) to maintain the farms in a certain moisture along time. The system can additionally record the daily temperatures to enable the farmers for selecting the suitable plants that are appropriate to each particular land. As a result, this automation system increases the productivity of the crops and optimizes the use of water supply. Mohanraj & Kirthika Ashokumar (2016) have proposed an e-Agriculture apps based on knowledge and monitoring models to enhance the water utilization and reduce the labor cost. The knowledge model is built based on some existed crop information data however the monitoring model is built using TI CC3200 Launchpad sensors modules, which are together enabled to overcome the limitations of manual system in the water and labor utilization.

Prathibha, Hongal & Jyothi (2017) have developed temperature and humidity monitoring in agricultural products using CC3200 single chip sensor. A camera is connected to CC3200 for capturing live stream video which is sent to the smartphone of farmers using Wi-Fi, in order to apply the suitable manual action. On the other side, Kajol, Akshay & Keerthan (2018) presented an IoT based monitoring system for agricultural field analysis with capability for advising the farmers on the soil status moisture, pest treatments and selection of the suitable crop to the soil. In such a system, a robot with a Raspberry microcontroller is able to drive in the field and monitor the moisture of soil in a distance of 100 m which is saved in clouding file for further processing. Upon completing the moving in the field, the storage data in the clouds are reported and sent to the farmer. Athawale et al. (2020) have proposed an IoT based smart watering system using smartphone to update the farmer on the garden watering status. It implements a soil sensor together with ESP8266 microcontroller to monitor the plant and enable the farmer to supply the water.

A smart agricultural monitoring system based on IoT, and sensors technologies has been used to improve the farm productivity, quality and response to the climate change (AshifuddinMondal & Rehena, 2018). The soil’s humidity and temperature sensors are used to monitor the soil status where their data are used to apply the necessary action and stored in database, called Thing-speak cloud, for future analysis. A monitoring system that can help in taking care for horticulture farming is developed by Kaburuan, Jayadi & Harisno (2019). Two types of data are used in this system, namely, data collected by IoT with sensors and database of daily weather at the cultivation places. The fusion of such data is used to train AI model that enables for building an automated climate horticulture monitoring system. An automatic monitoring of green house and pH level during a hydroponics planting, is presented in Saraswathi et al. (2018). Several sensors are used for measuring the temperature, pressure, and humidity quantities of the plants inside the green house and supplying the solution of nutrients, if needed. IoT is used to send the data to the user apps for monitoring and continuous checking of the plants.

An automatic watering system is used for controlling the water flow in the rice plants (Rani, Kumar & Bhushan, 2019). A kind of irrigation planning has been proposed based on monitoring the soil, weather and crop. The pH level, temperature and moisture sensors are used to detect the need of the water in the soil, which is automatically supplied using water pumps. The data are transferred through IoT operating system with a dashboard and http protocol that can enable the user to turn water pumps on or off as needed. An AI-based automated plant monitoring is used for supplying the water into the soil based on sensors and IoT data (Ahmad et al., 2021). Three types of sensors, namely, temperature, humidity and soil moisture have been used to detect the soil status which is transferred through Node-MCU ESP32 to mobile application using cloud-based Blynk and IoT analytics with Thing-Speak tools. The water is controlled using fuzzy logic algorithm to reduce the losses and increase efficiency. Kishore, Kumar & Murthy (2017) have supplied the waters to the plant based on the reading of the soil moisture sensor and Arduino interfaced system that can detect the signs of diseases. A BMP 180 sensor for measuring the temperature and pressure values in the surrounding environment, is used to trigger a heater when the temperature falls within specific cooling conditions. The image processing in such system is accomplished based on the Kekre transform and variance methods.

Chowdhury et al. (2021) proposed an efficient net base deep learning algorithm for segmentation and classification of tomato leaves diseases. Two models, namely, U-net and Modified U-net are used for segmentation with binary classification into healthy and unhealthy conditions. Sathiya, Josephine & Jeyabalaraja (2022) have developed an automatic AI-visual system for herbs plant classification and disease detection using hybrid of multi-swarm coyote optimization and improved Chan-Vese snake optimization algorithms. The former is used for segmenting the disease into images through recognizing the area of diseases among of the plant leaves image. The latter is used to reduce the features required for leaves classification. A deep learning classifier is used for classifying the herbs leaves into Healthy, gray mold, rust and scab diseases.

This article is aimed to develop an automated AI-powered IoT plant monitoring systems to increase the efficiency of planting, eliminate the noise that can cause wrong decisions and enable the farmers to track the plant condition. The proposed system has been applied on chili plant due to the following reasons:

- Chili is very sensitive to the soil moisture, weather changes and can be affected with lack of monitoring (Husin et al., 2012).

- It can be attacked by several types of diseases caused by bacteria and fungi as shown in Table 1.

- The life cycle of chili is too short, so if the dying of the plant leaves is happened, it is difficult to survive and get good yields after that.

Table 1 Common diseases for chilli plant (Chowdhury et al., 2021).

Category	Name	Symptoms	
Bacterial	Bacterial leaf spot	Water-soaked lesions that dry out and turn brown forming on the underside of the leave and Cracked brown lesions on fruit	
Fungal	Cercospora leaf spot	Circular spots with brown margins and grey centre appear on leaves and the spots enlarge and coalesce with others	
Fungal	Alternaria leaf spot	Spots enlarge, and by the time they are one-fourth inch in diameter or larger, concentric rings in a bull’s eye pattern can be seen in the centre of the diseased area and Tissue surrounding the spots may turn yellow hole	

Table 1 shows the common diseases that affect the chili plants.

The main challenges in this project are related to the data processing of sensors where each sensor data has been lonely processed with a lot of noise. Thus, a novel AI algorithm has been introduced to eliminate the noise and take the right decisions, which is challenging and complex mathematical task. Other challenges are:

- Due to limited budget, a small prototype has been built to test the proposed system.

- The selected sensors are cheap and noisy which is resulted in a need to develop a robust AI based-noise elimination algorithm.

- The study is applied on chili plant as a case study which is very sensitive to the soil moisture, weather changes and can be attacked by several types of diseases.

The contributions of this work can be summarized as follows:

- A complete solution for automatizing the process of Chili plant monitoring and actuating, has been introduced in this article. In fact, the current technology has not been yet completely solved the detection of crops dying and unwanted yield (Abdullah et al., 2021).

- The key contribution of this work relies on developing of a new AI based noise elimination in post-image processing to distinguish between the spots caused by noise and real pests.

- An IoT system integrated with the proposed AI, is used to send a report on the crop conditions and the action that have been made to the farmers through IoT, to enable them for tracking the crops and ensure the plant’s healthiness.

- A small scale prototype has been developed to test the proposed AI-powered IoT algorithm for monitoring and taking the right action on the chili plant.

Methodology

This work passes through several methodology’s activities to come out with the prototype of automated IoT monitoring system as follows:

Design of the measurement and monitoring system

The greenhouse is a special planting structure that can generate suitable conditions for the plants, enabling the plants to grow from sowing until harvesting. The main challenge in the greenhouse is how to isolate it from outside weather by selecting good greenhouse materials. For the weather inside the greenhouse, one can use the sensors to measure the plant’s surrounding conditions and apply the necessarily actions either by people in the manual methods or by actuators in automatic farming. Such actions will be applied based on the type of plants and recommendations by agricultural engineer. Thus, the practical challenges related to the variability in outside environmental conditions, geographic region and plant responses due to outside-weather, have no considerable effects on the planting processes in the greenhouse.

In the concept design of the proposed system as shown in Fig. 1A, three main different subsystems have been used, namely, irrigation, CO2 detecting and plant infects detection systems, to simultaneously work with microcontroller and IoT systems.

Figure 1 (A and B) Concept design of the system: proposed sub-systems of irrigation system, CO2 detecting system and plant infects detection system.

In the irrigation system, a soil moisture sensor is immersed in the soil to detect the moisture level of the soil surrounding the plants and send the signals to the microcontroller. If the soil moisture level is low, the microcontroller will turn on the pump 1 and the water will be pumped into the soil. With CO2 detection system, the CO2 level of the greenhouse environment is measured every minute and if the CO2 level around the plant is high, the microcontroller will turn on the exhaust fan to circulate the air inside the greenhouse environment. Therefore, the CO2 level in the air will return to its normal level again. In the plant infects detection system, a camera installed on top of the plant, is used to give a live steaming video of the plant. If the plant is found to be infected by pests, pesticide will be pumped to the plant using pump 2.

This concept of design was generated based on three main aspects, namely, sensors, actuators and data processing. Five sensors are used to obtain the measurements for each system, such as soil moisture, humidity and temperature detection, CO2 detector and camera sensors. The system applies the necessary actions on the plant using actuators such as irrigation water pumps, fans and LED lamps. The data processing of the system includes the signal and image processing of sensors reading and IoT system. In this design, the camera detects the conditions of the plant’s leaves through a live stream video which is processed in MATLAB using image acquisition and image processing toolbox. The soil moisture sensor is used to detect the moisture level of the soil of the chili plant. The sensor signals are then processed in the embedded system and the results are sent to the irrigation pump. The CO2 detector is used to detect the concentration of CO2 levels in the air inside the greenhouse. When CO2 concentration is high, the exhaust fan is used to circulate the air inside the greenhouse and reduce the concentration of CO2.

The air humidity sensor is able to detect the humidity, as well as the temperature of the air in the greenhouse, so that enable to determine if the condition is suitable to the plant. The relays are used to automatically turn on/off the power supply to the pumps and fans, according to the data coming from microcontroller of the proposed system.

Figure 1 shows the concept design of the proposed system.

The electronic breadboard is used to create a tidy and well managed circuit as shown in Fig. 1B. Firstly, 5V and 3.3V (VCC) power supply from the Arduino microcontroller is connected to the top and bottom positive terminal respectively. Then, the ground terminal is connected to both negative terminals to complete the circuit. The VCC terminal of the relay is connected to the 5V circuit while the VCC terminals of the soil moisture, CO2 detector and temperature sensors are connected to the 3.3V circuit. Next, the ground terminal from either side of the 5V and 3.3V circuit is connected to a new column of the breadboard. The ground terminals from all sensors and relays are then connected to this column. For data input, A0 terminal from the soil moisture sensor is connected to the A0 terminal of Arduino, A0 terminal of CO2 detector sensor is connected to the A1 terminal of Arduino and the OUT terminal of temperature sensor is connected to the D7 terminal of Arduino. The inputs of relay’s channel 1 and channel 2 are connected to the D8 and D9 terminal of Arduino, respectively. For the actuators, both positive terminals of the batteries are connected respectively to the Common (C) terminal of the relay channel 1 and channel 2. Then, the positive terminal of the pump is connected to the Normally Closed (NC) terminal in channel 1 while the negative terminal is connected to the negative terminal of the battery 1 to complete the circuit. Lastly, the positive terminal of the exhaust fan is connected to the Normally Opened (NO) terminal in channel 2 while the negative terminal is connected to the negative terminal of the battery 2.

Development of monitoring system prototype

A small greenhouse prototype with dimensions of 70 × 50 × 50 cm has been developed in the lab to test the proposed system as shown in Fig. 2. A waterproof box with a plastic cover are placed at the sides of the plant to seal the greenhouse and protect the electrical components and circuit from the surrounding environments. A webcam called, 360 Degree with Microphone, is located at one corner of the box for stream live recording of the plant status. Four sensors, namely, moisture sensor module (Model: SN-MOISTURE-MOD), CO2 detector sensor module (Model: SN-MQ135-MOD), temperature and humidity sensor (Model: DHT22) are put in the plant soil, beside the plant and inside the green house, respectively. The fan with DC5V Brushless Fan 6,000 rpm-Jetson Nano model is used for circulating the air inside greenhouse. Two Micro-Submersible Water Pumps DC 3V–5V are used to supply water and apply the suitable treatment of the pest. In addition, Arduino with a model of MKR WiFi 1000, is used as microcontroller to receive data from sensors, send data to actuators and processing data the data in C-language program. 2 Channel 5V Active Low Relay Module (Model: BB-RELAY-5V-02) is used to switch the pump and fan with the data coming from microcontroller. The total cost of the proposed system is around 91.8USD.

Figure 2 Greenhouse prototype used in the proposed system.

The system is built in the lab with a small scale greenhouse prototype using cheap sensors, actuators and microcontroller. It can be enlarged for big scale environments such the standard greenhouse with dimensions of 2,000 × 900 × 300 cm, by increasing the number of sensors, actuators and micro-controllers in three rows arrangements: two on the sides of greenhouse and another one on the ceiling. The data processing can be done in separate microcontrollers and sent the decisions to farmer through IoT. Such system is cost-effective as the capital cost of the system may be higher at the beginning comparing with the manual system but the operational cost is lower. The most important feature of the proposed system is that it can discover the diseases that harm the plant once it happens which cannot be detected by manual methods with operators.

The proposed system can be used in the greenhouse for any kind of plants by utilizing the sensors to adjust the weather conditions inside the greenhouse based on the plants standards and recommendations of agricultural engineer. We have applied the proposed system on the chili plant as it is effected too much by the weather changing.

In other words, the proposed system can be applied to any other plant with changing of the threshold of the sensors during processing, such as temperature, humidity and CO2 level, so that can control all the above-mentioned parameters based on the required conditions. In fact, the sensors will keep updating on the status of plant conditions and the decision making will be taken by signal processing and ELSL algorithms which is then converted into electrical signals to turn on/off pumps and fans.

Sensors’ signal and image processing

Two programming languages, namely, Arduino IDE and MATLAB are used in this project for signal and image processing, respectively. The algorithms used for both processes are analyzed in details as follows:

-Sensors signal processing

Three sensors are used to sense the soils moisture, CO2 level and humidity with temperature inside greenhouse. The signals of those sensors are processed in Arduino microcontroller to extract the suitable features as follows:

- Soil moisture sensor:

SN-MOISTURE-MOD sensor module is used to measure the moisture in the soil. If the moisture sensor output is low, then the pump will switch on, however, when the moisture sensor output is high, the pump will be switched off. The pseudo-code algorithm for signal processing of this sensors is illustrated bellow: → Assign the input of relay to channel 1.

→ Assign soil moisture sensor to analog input pin.

→ Establish Setup function () for reading. Setup baud rate into 9600

Read serial port of sensor.

Switch relay-pin into out mode

Switch sensor moisture into input mode

End function setup

→ Establish loop function () for reading. Create Variable soilmoisturePin to read the data of analog port analogRead() function

Use map function () inside loop function. Display the moisture level.

If statement (moisture level is low) then relay channel 1 will stay at the NC channel, and the pump is turned on.

else the relay channel 1 will switch to NO channel, and the pump is turned off.

End if statement

End map function

→ End loop function

- Gas sensor

The MQ-135 gas sensor is used to measure the level of CO2 inside the greenhouse. If the gas conductivity is low (less than 50 ppm), then the exhausted fan will switch off, however, when the gas conductivity of gas sensor is low, the fan will be switched off. The pseudo-code algorithm for signal processing of this sensors is illustrated bellow: → Assign the input of relay to channel 2.

→ Assign MQ-135 to analog input pin.

→ Establish Setup function () for reading. Setup baud rate into 9600

Read serial port of gas sensor.

Switch relay-pin into output mode

Switch sensor moisture into input mode

End function setup

→ Establish loop function () for reading. Calculate conductivity in percents

Create Variable gasPin to read the data of analog port analogRead() function

Use map function () inside loop function. Display the gas level.

If statement (conductivity <50) Then relay channel 2 will stay at the NC channel, and the exhausted fan is turned off.

else the relay channel 2 will switch to NO channel, and the exhausted fan is turned on.

End if

End map function

→ End loop function

- Temperature and humidity sensor

DHT22 sensor is used to measure the humidity and temperature of greenhouse. It enables to track the temperature inside the greenhouse and check whether the condition is healthy for the plant or not. The pseudo-code the algorithm is described herewith: → Include DHT.h library.

→ define DHTPIN pin to indicate the location of the sensor input pin

→ define DHTTYPE DHT22 to indicate the type of DHT sensor in our system

→ Initialise DHT sensor for normal 16mhz Arduino

→ Define variables humidity and temperature as float variable

→ Establish loop function () for reading. Create Variable soilmoisturePin to read the data of analog port analogRead() function

Use map function () inside loop function. Display the moisture level.

If statement (moisture level is low) then relay channel 1 will stay at the NC channel, and the pump is turned on.

else the relay channel 1 will switch to NO channel, and the pump is turned off.

End if statement

End map function

→ End loop function

Video and image processing

The video and image processing are accomplished in MATLAB as it has ready image processing toolboxes. Three main stages are used to detect the pest and defects on the leaves of chili as follows:

Video pre-processing

A live streaming video is used to detect the pests on the leaves which is decomposed into 60 frames per second. A frame per second is used for images analysis and features detection. The image pre-processing stage is used to prepare the frame in a good format for further processing in the processing and post-processing stages. It includes operations that can adjust the brightness, crop certain part of image and converting into grey scale. The results of the algorithm pre-processing processes are shown in Fig. 3. The Pseudo-code of image pre-processing algorithm is described herewith: → Construct an input video object using vid=videoinput(’winvideo’,1);

→ Preview the constructed video object: preview(vid);

→ Start decomposing of the video into frames: start(vid)

→ Take one frame per second from 60frame/s: input=getdata(vid,1);

→ imcrop() function to crop the image into a desired size: GH=imcrop(input,[x1 y1 x2 y2]);

→ rgb2gray() function to convert into grayscale : GS=rgb2gray(GH)

Figure 3 Video pre-processing algorithm: (A) Display of the analyzed frame image. (B) Display of cropped image GH. (C) Display of grayscale image GS.

Image processing

Several operations will be applied on the frame that has already passed the pre-processing stage. This stage is aimed to find the features of the plant such as pests dark spots, holes and change of color on the leaves, which will be analyzed later to determine whether it is a pest or noise on the leaves. For this purpose, multiple filters are used for blurring image and edge detection using Gaussian filter and Canny filters, respectively as shown in Fig. 4. Canny filter has shown better performance in comparison with Prewitt and Sobel filters as shown in Figs. 4D–4F. The Pseudo-code of image processing algorithm is illustrated herewith: → Apply blurring to the image using Gaussian filter: PSF=fspecial(’gaussian’,5,8);I=imfilter(GS,PSF,"symmetric","conv");

→ Apply edge detection methods using Sobel, Prewitt and Canny: BWS=edge(I,’sobel’); BWP=edge(I,’prewitt’); BWC=edge(I,’canny’);

→ Choose suitable edge detection

→ Fill up the dark patches or holes in the image with white colour using the imfill() function P = imfill(BWC, ’holes’);

→ Invert black and white colour of image P: P= ~P.

Figure 4 Image processing algorithm results.

The results of image processing algorithm are shown in Fig. 4.

Figure 4 shows that the pest spots on the leaves from the image are normally occurred as big dark holes. However, it is mixed with noise, which have to be eliminated using AI algorithms.

Noise elimination in signals and images

After applying the image and signal processing, there still some noise that must be eliminated to make the right decision and apply the necessary actions. The signals of moisture, CO2 emission and temperature sensors includes some wrong reading due to the sensor quality and unexpected condition. This include two main noise: (1) the missing data and (2) the data that are too high or too low than normal measurements. The result of image processing includes many noise that are looked similar to the defects on the leave. These noise are not avoidable and coming from many sources such as: (1) light homogeneity (2) background of the image (3) unexpected object (4) shadow that may occur during capturing the live video.

Thus, there is a need for AI algorithm to classify the spots in the leave into defective and non-defective areas.

Signal based noise elimination algorithm

For the signals that acquired by moisture, CO2 emission and temperature sensors, several types of pass filters are used to remove the unwanted signals which are too far from the range of values. This is done by determining the threshold upper and lower level for each sensor. The low pass filter is used for moisture sensor measurement as in Eq. (1), however high pass filter and band pass filters are used for CO2 emission as in Eq. (2) and temperature sensors as in Eq. (3), respectively.

(1) yns={x1x1≤20%0x1>20%}

where x1 is the reading values of the moisture sensor, yns is the moisture sensor measurements after applying the low pass filter.

(2) yco2={x2x2≥50ppm0x2<50 ppm}

where x2 is the reading values of the CO2 emission detector sensors, yco2 is the emission detector of CO2 after applying high pass filter

(3) yts={0x3<12C0x312C0≤x3≤15C00x3>15C0}

where x3 is the reading values of the temperature sensor, yts is the temperature sensor measurement after applying band pass filter

Image-based noise elimination algorithm

Since, there are a lot of noise in the images after post-processing, a new AI algorithm called enhanced laser simulator (ELSL) has been introduced here to accurately find the defected areas of the leaves.

Principle of Enhanced laser Simulator Logic (ELSL):

It is an extension of Laser Simulator Logic (LSL) (Ali et al., 2022) that has been used for noise reduction. Both LSL and ELSL are capable for handling noise through high inference of linguistic variables with a dynamic range of membership functions. Such accommodation of the noise enables to avoid the drawbacks of fuzzy logic membership functions when it is dealing with highly overlapping membership functions of the linguistic variables, as shown in Fig. 5. Equations (4)–(13) present the fuzzification process in ELSL with lowly and highly inferenced linguistic variables. In comparison with LSL, ELSL simplifies the computation of membership functions for high inference linguistic variables, as derived in Eqs. (11)–(14). The main features of ELSL comparing with LSL are listed herewith:

- A step function to delete the effect of non-occurrence of linguistic variable is newly proposed, as given in Eqs. (11) and (12).

- A step function for calculation of any lowly or highly overlapped membership function of the crisp input is introduced in Eq. (13).

- A general equation that simplifies the calculation of membership values for both low and high overlapping situations is formulated in Eq. (10).

Figure 5 Fuzzy logic drawbacks with high overlapping.

LSL approach depends on finding the ratio between the crisp values location and the universe of discourse of linguistic variables, i.e., the membership value of the crisp input is computed as the division of its value position into the whole input linguistic variable range as described in Eq. (4). The membership function of the crisp input/output in LSL and ELSL is modeled as drawn in Fig. 6. The membership values of inputs linguistic variables are then implicated on the corresponding output linguistic variables, according to fuzzy inference rules, to determine the output membership values.

Figure 6 Inference system for a single linguistic variable in laser simulator logic.

Let us define the membership value of a fuzzy input/output set (Q) in the universe of discourse (X) as: μi:X→[01] with Q={(x,μi(x))|x∈X}. The ELSL membership values μi(x) are simply presented in Fig. 6 and computed using Eq. (4):

(4) μ(x)={0x≤sx−a(x−a)+(m−x)s<x<mb−y(b−x)+(x−m)m<x<e0x≥e}

where s, e and m are the starting, ending and midst of linguistic variable as illustrated in Fig. 6.

For x = y1 and x = y2 as shown in Fig. 6, the memberships are:

μ(y1)=y1−s(y1−s)+(m−y1),μ(y2)=e−y2(e−y2)+(y2−m)

The accumulative membership values for a specific crisp input across all linguistic variables has to verify Eq. (5):

(5) ∑i=1j⁡μi(x)≤1|x∈X

where j is the number of inferenced linguistic variables.

ELSL derivation

Let us consider that there are j inferenced-linguistic variables in the input set with universe of discourse X. The inferenced-linguistic variables have the following starting, midst and ending points as depicted in Fig. 7:

- s1, s2, s3………, sj are the beginning values of the ranges in the linguistic variables

- m1, m2, m3 ……..…., mj are the midst points of the ranges in the linguistic variables

- e1, e2, e3……..…, ej are the utmost values of the ranges in the linguistic variables

Figure 7 Types of overlapping in membership functions for triangular fuzzy logic and laser simulator logic.

It is required to formulate a general equation that can be used to calculate the membership values in both low and high overlapping scenarios. To achieve this, Eqs. (4) and (6) are derived to calculate the membership values and the accumulative membership value in lowly overlapping situations for a specific crisp input, respectively.

(6) ∑i=1jμi(y)={0x<s1x−a1(x−a1)+(m1−x)s1<x<m1e1−y(e1−x)+(e1−m1)+x−a2(x−s2)+(m2−x)+…+x−sj(x−sj)+(mj−x)m1<x<e1e2−x(e2−x)+(e2−m2)+x−s3(x−a3)+(m3−x)+…+x−sj(x−aj)+(mj−x)e1<x<m2⋮ej−1−x(ej−1−y)+(ej−1−mj−1)+x−sj(y−sj)+(mj−y)mj−1<x<ej−1ej−x(ej−x)+(x−mj)mj<x<ej0x>ej}

The accumulative membership values for highly overlapping case can be written as in Eq. (7):

(7) ∑i=1jμi(x)={0x≤s1x−s1(x−s1)+(m1−x)s1<x≤s2x−s1ΔD+x−s2ΔD+…+x−sjΔDa2<x≤m1e1−xΔD+x−s2ΔD+…+x−sjΔDm1<x≤e1&x≤m2e1−xΔD+e2−xΔD+…+x−sjΔDm1<x≤e1&x≥m2e2−xΔD+x−s3ΔD+…+x−sjΔDe1<x≤e2&x≤m3e2−xΔD+e3−xΔD+…+x−sjΔDe1<x≤e2&x≥m3⋮ej−1−xΔD+x−sjΔDej−2<x≤ej−1&x≤mjej−1−xΔD+x−sjΔDej−2<x≤ej−1&x≥mjei−x(ei−x)+(x−mi)ej−1<x≤ej0x≥ej}

where ΔD in Eq. (7), has the values as in Eq. (8):

(8) ΔD=∑i=1jΔDi={0x≤s1(x−s1)+(x−m1)s1<x≤s2(x−s1)+(x−s2)+…+(x−sj)s2<x≤m1(e1−x)+(x−s2)+…+(x−sj)m1<x≤e1(e2−x)+(x−s3)+…+(x−sj)s1<x≤e2⋮(ej−1−x)+(x−sj)ej−2<x≤ej−1(ej−x)+(x−mj)ej−1<x≤ej0x≥ej}.

To address well ΔD sub-ranges in the Eqs. (7) and (8), they have been implicitly modified with Eq. (9), to determine the membership value as a minimum sub-range measured from the midst point to the starting and ending points in each respected linguistic variable. This allows to compute the membership values at any location, no matter it is located before or after the midst point of the linguistic variable, as given in Eq. (9):

(9) min((x−si),(ei−x))={x−sisi<x<miei−xmi<x<ei}.

Thus, ΔD in Eq. (8) is written as Eq. (10) using the sub-ranges defined in Eq. (9):

(10) ΔD=∑i=1j⁡ΔDi={∑i=1j⁡min((x−si),(ei−x))s2<x≤e1∑i=2j⁡min((x−si),(ei−x))e1<x≤e2⋮∑i=j−1j⁡min((x−si),(ei−x))ej−2<x≤ej−1}.

To deal with the cases where the beginning and ending ranges points are not occurred in the order, a step function fi is introduced to eliminate the effect of the linguistic variables when they are already exceeded or still not reached yet. fi is given in Eq. (11):

(11) fi={1min((x−si),(ei−x)≥00min((x−si),(ei−x)<0}.

To nullify the effect of the non-occurrence linguistic variables in ΔD, Eq. (12) is used:

(12) fi×min((x−si),(ei−x))={min((x−si),(ei−x))x≥si0x<si}.

To find the membership value of a crisp input, regardless they are lowly or highly overlapped, the expression |(mi−x)| in Eq. (4) has to be involved in Eq. (10) when there exists a low inferencing of linguistic variable. However, it has to be excluded when the high inferencing is occurred. This task can be accomplished using a step function Nx in Eq. (13):

(13) Nx={1x≤s1∨x≥ej−10s1<x<ej−1}.

The general equation for calculating the membership values of both low and high inference linguistic variables is written in Eq. (14).

(14) μi(x)={0x<si∨x>eimin((x−si),(ei−x)[∑i=i−11fi×min((x−si),(ei−x))]+|(mi−x)|×Nxsi≤x≤ei}.

Implication and Defuzzification of ELSL:

In the implication process, the membership values of the input linguistic variables are implicated on the corresponding output linguistic variables, according to fuzzy inference rules, to determine the output membership values. The implication values in ELSL is calculated using Eq. (15).

(15) zi=μi(x)×Zi

where Zi is the output linguistic variable range, zi is the result of input implication on the corresponding output. The crisp output is computed as an average value of the implicated values of zi in all rules using Eq. (16):

(16) z=∑i=1n⁡zin

where n is the number of implicated rules on the corresponding output linguistic variables.

Implementation of the proposed AI

To identify whether the spots in Fig. 4F are corresponding to diseases or noise, we use two input sets:

1st one is the different of intensity between the spots and the average of leave color intensity. If the different is high, it means that this is possible of defects. This is due to that the color of the disease spot is has intensity which looks totally different than other spots that maybe occur due to the shortage in chlorophyll. If the different is high, it means that this is possible of defects.

Intensity = {low, medium, high}

To accomplish the calculation of intensity, the following steps must be done:

(a) The location of the spots is determined from the result of image processing as in Fig. 4F.

(b) We should find the intensity of these spot areas in the original image as shown in Fig. 3B.

(c) The average of the original image intensity is calculated from image in Fig. 3B.

(d) The difference in intensity between step (b) and (c) is considered as the crisp input for intensity input set.

The 2nd Input set is the area of spots, where the big areas are possible to be defects:

Area = {small, Medium, big}.

The area is calculated for the spots with black pixels in the result of image processing algorithm as shown in Fig. 4F, which will be supplied as the crisp input for Area input set.

Figure 8 shows the input sets of ELSL.

Figure 8 ELSL input sets.

The output set of ELSL is the classification of the tested spots state into a defective or non-defective spots. In fact, we preferred to select binary decisions for the spots either defect or noise, as if we increase the linguistic variables of the output to include more situations such as possible noise and possible defects, it is required during defuzzification process to grab these new linguistic variables (possible noise and possible defect) into either noise or defect decisions, respectively:

Spot State = {Defect, Noise}.

The ranges of linguistic variables in the input and output sets are given in Table 2. The ranges have been selected based on empirical trials for some spots that we know its state whether defect or noise. The selected ranges cover all possibility cases.

Table 2 Inputs and outputs linguistic variables ranges in ILSL.

	Input set 1: Intensity (ratio 0–1)	Input set 2: Area (pixels)	Output: spot state (ratio 0–100)	
Linguistic variable 1	0–0.45 (Low)	0–2,000 (Low)	0–60 (Noise)	
Linguistic variable 2	0.2–0.8 (Medium)	800–5,000 (Medium)	40–100 (Defect)	
Linguistic variable 3	0.55–1 (High)	3,200–12,000 (High)		

The membership functions in ELSL is shown in Fig. 9.

Figure 9 ELSL membership input and output functions.

The rules for implicating the input on the output are listed as follows: 1) If Intensity is Low and Area is Small THEN Spot_area is Noise

2) If Intensity is Low and Area is Big THEN Spot_area is Noise

3) If Intensity is Medium and Area is Small THEN Spot_area is Noise

4) If Intensity is Medium and area is Medium THEN Spot_area is Defect

5) If Intensity is high and Area is Medium THEN Spot_area is Defect

6) If Intensity is High and Area is High THEN Spot_area is Defect

The ELSL steps and procedures are illustrated in Fig. 10.

Figure 10 ELSL flow chart.

The decision of ELSL on the spot area is highlighted in red circle if it is defect and no color if there it is noise. Figure 11 shows the resulted image after implementing ELSL.

Figure 11 Postprocessing of image: (A) original cropped image; (B) image after image processing algorithm; (C) image after implementing ELSL algorithm.

The final results of video pre-processing, processing and post-processing with ELSL algorithms are shown in Fig. 12. When the red circles are detected on the leaves, the pump will spray the pesticide on the leaves, however, due to the effect of such pesticide to the people in lab, we, instead, spray water to the leaves.

Figure 12 Results of video processing algorithm: (A) one frame from video (B) image after post-processing using ELSL.

Results and discussion

The proposed system consists of four main sub-systems, namely, the irrigation system, CO2 detecting system, temperature & humidity system and image processing system. The irrigation system, CO2 detecting system and temperature & humidity system are programmed in Arduino IDE-supported IoT while the image processing system is programmed in MATLAB. The results are reported as follows:

IoT signal processing results

In this project, it is intended to improve the automated plant monitoring system using the IoT system, which will enable the farmer to know the status of the plant in the farm without physical visit.

From Fig. 13, it can be clearly seen that all the readings obtained from the sensors are shown in the monitor of Arduino IDE. The readings detection of the monitoring system is repeated every second.

Figure 13 Continuous monitoring of system with Arduino MKR1000 readings every second.

By connecting the Arduino MKR WiFi 1000 to WiFi, one can monitor the reading of Arduino through IoT. For this purpose, we created a dashboard on IoT based Arduino IDE that can display all the readings (soil moisture level, CO2 level, temperature, air humidity) from each sensor as shown in Fig. 14. The display of the sensor is updated every second.

Figure 14 Dashboard of Arduino IoT system on laptop.

By using the Arduino IDE IoT system, one can also view the dashboard with live reading on the phone. An app named IoT Remote is installed on the phone to make it easier for farmer to monitor the farm from his place as shown in Fig. 15.

Figure 15 Dashboard of IoT remote app for mobile phone.

In addition, we have tested the proposed automated irrigation system by measuring the soil moisture level and actuating the water pump. When the detected soil moisture level is less than 20%, the pump will be turned on to supply the water into the plant as shown in Fig. 16A. The water supply will continue until the amount of the water nearby the plant reaches a certain level (above than 20%) in the greenhouse and the reading of the soil moisture level will become high. As a result, the pump will be turned off as shown in Fig. 16B. The pump will be turned on again once the moisture level of the soil becomes lower than 20%.

Figure 16 Response to the soil moisture sensor: (A) Water is pumped out when the moisture sensor is less than 20%. (B) No water is pumped out when the water is above than 20%.

For the CO2 detecting system, the exhaust fan is switched off when CO2 level is below than 50 ppm, as shown in Fig. 17. In fact, it is hard to obtain a CO2 level that is larger than 50% when there is only one plant in the green house. So, we used another way to get a value above than 50% as shown in Fig. 18. In this way, a lighter close to the MQ-135 gas sensor has been used to increase the CO2 level into above than 50%. The exhaust fan is then switched on to circulate the air inside the greenhouse environment. When the CO2 level is dropped below 50%, the exhaust fan is switched off again.

Figure 17 CO level is below than 50%: (A) Low CO2 level. (B) Fan is turned off.

Figure 18 CO level is above than 50 ppm: (A) Lighter is close to the MQ-135 sensor. (B) Fan is turned on (C) High CO2 level.

IoT based video processing results

A live IoT based video processing is used to continuously monitor the plant conditions and enable the farmer for tracking the health of leaves. In Fig. 19, we can clearly see that there are no pests on leaves; thus, no circles marked on the defect-less leaves. However, when the video processing is applied on the defective plant’s leaves with pests, it is occurred in the post-image processing as darker holes which is then marked by red circles as shown in Fig. 20.

Figure 19 Video processing on defect-less leaves.

Figure 20 Video processing on leaves with defects.

By repeating the video processing many times, we can conclude that the accuracy of the image processing system is around 75% as shown in Fig. 21 and Table 3. The farmer will be able to track the video of plant and the final results of image inspection with necessary decision through IoT connection.

Figure 21 Video processing of several types of leaves: (A) image processing on defectless leaves, (B) image processing on leaves with defects, (C) image processing on leaves with defects.

Table 3 A total of 20 trials for testing the video processing algorithm.

Trials	1	2	3	4	5	6	7	8	9	10	11	12	13	14	15	16	17	18	19	20	
Real pests	0	1	1	1	0	1	1	1	1	0	1	0	1	1	1	1	1	1	0	1	
Detected pests	1	1	1	1	1	1	1	0	1	0	1	1	1	1	1	1	0	1	0	1	

Table 3 shows the results of 20 trials for pests’ detection using video processing. The presenting of the pests on the leaves is marked as 1, however the pests-free is marked as 0. The comparison is done between the real pests as in 2nd row and the detected pests using video processing as in 3rd row. When there is an agreement between the real pests and video processing pests such as both have 0 or 1, the result is considered as successful detection. However, when they have different values, (e.g., if real pest is 1 and video processing pests is 0, it is resulted by wrong detection). The decision will be taken based on a comparison between the agreement detection of pests in both real and video processing pests even-though the number of the darker holes will not be same in both real and video processing pests as shown in Fig. 18C where the number of real pest’s darked holes (by eyes) are 16 however, the number of detected holes in the video processing (circled by red color) are 20. The results show that the algorithm is capable to detect the pests on the leaves with 75% successful rate.

Table 4 shows a comparison between the proposed system and the works presented in Gultom et al. (2017) and Husin et al. (2012) to monitor the chili plant. It is clear that the project in Husin et al. (2012) is not effective as it depends on the color feature, it is subjected to a lot of noise such as light non-homogenous and the changes of the colors due to of watering system. Also, such system can’t detect the source of the risks that are happened to the chili plant as there are no sensor to measure soil humidity, temperature and CO2 level. In addition to that, no IoT is used to inform the farmers. For the work that has been accomplished in Gultom et al. (2017) even-though it uses an automated actuating IoT system, but it lacks to inspect the chili leaves against the pests and doesn’t involve the use AI algorithm for taking the right decision. As a result, one can notice that the proposed system utilized more techniques and features to robustly detect the defects in chili plant, which is resulted by enhancing the accuracy of the proposed system.

Table 4 Comparison between proposed system and the systems presented in Gultom et al. (2017) and Husin et al. (2012).

	Techniques	Features	AI-Output classification	Pros	Cons	Accuracy	
Gultom et al. (2017)	– A moisture sensor

– pH and EC sensors

– Arduino microcontroller

– Web App using Ethernet

	– Dryness of soil

– The acidity of chili plants

	No AI	– Multiple sensors are used

– Automatic actuating IoT pumping system

	– No detection for pests on leaves

– No AI

	61%	
Husin et al. (2012)	– Camera and image processing

	– Only color of the plant

	Risk and healthy	– Good model using deep learning algorithm

	– Dependency on the color feature

	53%	
Proposed system	– Humidity, temperature and CO2 level sensors AND signal processing,

– Camera, image processing

– AI-based decision making

– IoT technology

	– Dryness of soil,

– Ratio of CO2 level,

– Changes of temperature,

– Area of defect,

– Average and spot intensities of leaves

	Defect and noise	– Has all required sensors to monitor the plant.

– Introduce a new AI algorithm

– Automatic actuating IoT pumping system

		75%	

The advantages of the proposed system are: (1) The system can autonomously monitor the plant conditions (2) The system can discover if the plant is effected by certain diseases (3) The system can apply the proper actions to the plant like operating pumps or fans (4) The system can send report to the farmer through IoT.

Conclusions

A novel AI-powered IoT plant monitoring system has been developed in this article. The system was able to monitor the plant’s low moisture level, high CO2 concentration level, and pests’ detection using signal and image processing algorithm incorporated with a novel AI for noise elimination. A small prototype for the chili plant has been built to test the effectiveness of the proposed AI-powered IoT based monitoring system. It uses an Arduino microcontroller to interconnect multiple sensors such as soil moisture, humidity, temperature, CO2 detectors and camera sensors, and actuators such as irrigation water pumps, fans and LED lamps in a single system, through open-sourced software, Arduino IDE to ensure that the coding runs the system smoothly. The data processing of the system includes the signal and image processing of sensors reading and IoT system. A dashboard for IoT based Arduino IDE has been created to display all the readings of sensors. When the soil moisture level detected is less than 20%, the pump will be turned on to supply the water into the plant. For the CO2 detecting system, when CO2 level is below than 50%, the exhaust fan is switched off. Results of video processing-based pests’ detection show that the algorithm is capable to detect the pests on the leaves with 75% successful rate. The limitation and future scope of this study are related to three main points: (1) Due to a limited budget, a small prototype has been built to test the proposed system. As future work a large scale prototype should be considered to test its efficiency. (2) The selected sensors are cheap and noisy which is resulted in a need to develop a robust AI based-noise elimination algorithm. Compact sensors can be used instead to reduce the dependency on AI algorithm for noise elimination. (3) The study is applied on chili plant as a case study which is very sensitive to the soil moisture, weather changes and can be attacked by several types of diseases. Other plants should be tested by the proposed system in future.

Supplemental Information

Supplemental Information 1 Live video processing.

Supplemental Information 2 Aurdino_IOT.

Supplemental Information 3 Aurduino code.

Supplemental Information 4 Recorded Video Processing.

Supplemental Information 5 Laser Simulator Logic code.

Supplemental Information 6 Datasets for images.

Additional Information and Declarations

Competing Interests

Author Contributions

Data Availability

The authors declare that they have no competing interests.

Mohammed A. H. Ali conceived and designed the experiments, performed the experiments, analyzed the data, performed the computation work, prepared figures and/or tables, authored or reviewed drafts of the article, and approved the final draft.

Khaja Moiduddin analyzed the data, authored or reviewed drafts of the article, and approved the final draft.

Yusoff Nukman analyzed the data, authored or reviewed drafts of the article, and approved the final draft.

Bushroa Abd Razak performed the experiments, prepared figures and/or tables, and approved the final draft.

Mohamed K. Aboudaif analyzed the data, prepared figures and/or tables, and approved the final draft.

Muthuramalingam Thangaraj analyzed the data, prepared figures and/or tables, and approved the final draft.

The following information was supplied regarding data availability:

A sample of the live video processing using laser simulator logic and the code are available in the Supplemental Files.

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
