# Peer review of "An automated AI-powered IoT algorithm with data processing and noise elimination for plant monitoring and actuating"

_PeerJ Computer Science, doi:10.7717/peerj-cs.2448_

## Round 0.1 · original submission · Major Revisions

Thank you for submitting your article. Based on reviews' comments, your article has not yet been recommended for publication in its current form. However, we encourage you to address the concerns and criticisms of the reviewer and to resubmit your article once you have updated it accordingly. When submitting the revised version of your article, it will be better to address the following:

1. “AI” should be written with its long form in its first usage.
2. “CO2” should be correctly written.
3. “Equations (1-3)”, “Eqs. 4-13”, “Eq. 5”, “Eq. (9).”, and etc. should be correctly written. Referencing to equations are untidily written.
4. Explanation of the equations should be checked. All variables should be written in italic as in the equations. Their definitions and boundaries should be defined. Please use equation numbers for referencing the equations. Do not use "as", "below", “following” “as follows”, etc. Necessary references should also be provided for relevant equations.

Best wishes,

Reviewer 1 ·

Basic reporting

The paper presents an overly ambitious goal of developing an AI-powered IoT system capable of autonomously monitoring and managing plant conditions without human intervention. Such claims often overlook practical challenges in real-world agricultural settings, where variability in environmental conditions and plant responses can complicate automated decision-making.
While the system claims to monitor and actuate based on plant conditions, the practicality of deploying such a complex system in diverse agricultural environments remains questionable. The reliance on multiple sensors and AI algorithms may introduce complexities that could outweigh the benefits, especially in terms of cost-effectiveness and maintenance.

Experimental design

Highlighting a 75% success rate in pest detection may suggest cherry-picking of results. Such a figure needs to be contextualized with details on false positives, false negatives, and the system's overall reliability under varying conditions typical of agricultural settings.
The paper lacks discussion on the practical challenges encountered during the design, fabrication, and testing phases. Real-world implementation of AI algorithms in IoT systems often faces challenges such as data integration, sensor reliability, and algorithm robustness, which are crucial for long-term functionality.

Validity of the findings

There is limited discussion on the scalability and adaptability of the proposed system to different plant types and environmental conditions beyond chili plants. Agricultural IoT systems need to demonstrate versatility and reliability across various crops to be considered practical and impactful.
The paper does not provide sufficient validation of the proposed system across different environmental conditions or geographic regions. This raises concerns about the generalizability of the findings and applicability of the system beyond the specific conditions tested.

Additional comments

While the paper mentions stages of system development and testing, details on the AI algorithms, signal processing techniques, and the AI noise reduction method are insufficient for replication and critical evaluation by other researchers or practitioners in the field.

Reviewer 2 ·

Basic reporting

Figures must be more clear and elaborative.
Figure 2 and 9 must be labelled properly.
More explanation of Plant infection detection must be added.
More relevant references must be added.
Spelling mistakes, equation numbers & subscript should be taken care of like bellow, mean etc.
Key contributions and paper organization are missing.
Subsections are not properly marked.
Notation of IOT and IoT should be uniform.

Experimental design

Total cost of the set up must be specified.
Explain whether the proposed experimental design is generic or specific to chili.

Validity of the findings

Concept design of the proposed system as shown in fig1 should be more elaborative.
Justify how physical monitoring is not required as far as : turning on or off the pump and fans.
Highlight future scope and limitation of the study.

Additional comments

Check for formatting and grammatical errors.

Reviewer 3 ·

Basic reporting

The paper presents a plant monitoring system with artificial intelligence and Internet of Things capabilities.
The soil moisture, CO2 detector, temperature, and camera sensors have been used in the system.
The chili plant has been used to test the proposed system.
A novel artificial intelligence noise reduction technique has been implemented.
First, there are some problems in the paper in terms of writing and presentation:
Abbreviations should be given where they are first used and should be in a standard. For example; IoT, IOT, CO2,...
The units on the figures and tables should be given in proper.
Some figures are not well presented. For example, Figure 12 can be given in a table.
The equations should be written in a standard and should be numbered. For example, Eq. 13 can not be read.

Experimental design

The research falls within the scope of the journal.
The research question is moderately well-defined, why the chili plant was chosen and its requirements need to be given in detail.
The algorithms (for example, noise elimination algorithm) should be given with the flow diagrams, pseudo-codes are not in a standard presentation, they are just programming codes.

Validity of the findings

The results should be discussed in terms of Chili plant specifications. The given findings are discussed for all plants.
What are the advantages of the proposed system for chili plant farming?

Additional comments

All the questions above should be answered and the paper should be presented with this journal's standards and rules.
The advantages of the system should be compared to other systems. Because of the given explanations, the proposed IoT monitoring system has no novel aspects.

---

## Round 0.2 · Major Revisions

Dear authors,

Thank you for your revised paper. Although two reviewers think that your paper has been improved after additions and modifications, according to Reviewer 1, your paper needs revision and we encourage you to address the major concerns and criticisms of this reviewer and resubmit your article once you have updated it accordingly.

Best wishes,

Reviewer 1 ·

Basic reporting

While the paper demonstrates success in a controlled environment, there’s little discussion on the scalability of the system for large-scale farming operations. Large farms often face varied environmental conditions, and the proposed system’s ability to handle diverse climates, terrains, and plant species is not fully addressed.
The algorithm for pest detection has a success rate of 75%, which might be insufficient for practical, real-world applications. For a sensitive crop like chili, a higher accuracy is crucial, as missed detections could lead to severe infestations and crop losses.

Experimental design

The system mainly focuses on detecting low moisture, high CO2 levels, and pests. However, chili plants are susceptible to various diseases that may not manifest as moisture or pest issues. The absence of disease-specific monitoring weakens the overall effectiveness of the system.
The design of the system includes multiple sensors, cameras, and actuators, which could result in high manufacturing and maintenance costs. For many farmers, especially in less developed regions, such an elaborate setup may not be economically viable.
I suggest the following ref. to cite in literature review & compare results.

Validity of the findings

Although the system aims to eliminate human interaction, in practice, farming often requires nuanced human decision-making that AI may not fully replicate. Issues like false positives from sensor readings or the need for manual calibration could limit the system’s autonomous functionality.
Testing on just one type of crop (chili plants) limits the generalizability of the system. There’s no information on whether this AI-powered IoT setup would be effective across other crop types, soil conditions, or weather patterns. More diverse trials are needed to confirm its broader applicability

Reviewer 2 ·

Basic reporting

Manuscript is now accepted after incorporating corrections.

Experimental design

no comment

Validity of the findings

no comment

Additional comments

Accepted

Reviewer 3 ·

Basic reporting

The revised version of the study is well-written and well-organized.
The references are appropriate.
Figures and tables in the paper are related and explained in the relevant paragraph.

Experimental design

The research question has been well-defined.
The conclusions are consistent with the evidence and arguments presented and they address the main question posed.

Validity of the findings

The performance results are consistent. There are comparative results.

Additional comments

The paper can be accepted as it is.

---

## Round 0.3 · accepted · Accept

Dear authors,

Thank you for the revised paper. The reviewers accept your paper and I confirm that you have addressed the concerns and criticisms. Your paper now seems ready for publication.

Best wishes,

Reviewer 1 ·

Basic reporting

Seems good work

Experimental design

Acceptable

Validity of the findings

Acceptable

Additional comments

N/A

Reviewer 3 ·

Basic reporting

The paper is well-written.
The references are appropriate.

Experimental design

There is a contribution to the literature.
The performance results are consistent. There are comparative results.

Validity of the findings

The conclusions are consistent with the evidence and arguments presented and they address the main question posed.
Figures and tables in the paper are related and explained in the relevant paragraph.

Additional comments

The paper can be accepted as it is.